# Talking About Weight with Children: Associations with Parental Stigma, Bias, Attitudes, and Child Weight Status

**DOI:** 10.3390/nu17182920

**Published:** 2025-09-10

**Authors:** Anca Georgiana Ispas, Alina Ioana Forray, Alexandra Lacurezeanu, Dumitru Petreuș, Laura Ioana Gavrilaș, Răzvan Mircea Cherecheș

**Affiliations:** 1Department of Public Health, College of Political, Administrative and Communication Sciences, Babes-Bolyai University, 400376 Cluj-Napoca, Romania; georgiana.ispas@publichealth.ro (A.G.I.);; 2Discipline of Public Health and Management, Department of Community Medicine, “Iuliu Hațieganu” University of Medicine and Pharmacy, 400012 Cluj-Napoca, Romania; 3Faculty of Food Science and Technology, University of Agricultural Sciences and Veterinary Medicine, 400372 Cluj-Napoca, Romania; 4Asociația Wello, 400686 Cluj-Napoca, Romania; 5Department 2, Faculty of Nursing and Health Sciences, “Iuliu Hațieganu” University of Medicine and Pharmacy, 400012 Cluj-Napoca, Romania

**Keywords:** parental weight stigma, internalized weight bias, antifat attitudes, eating behaviors, body image, parent–child communication, childhood obesity, stigma reduction, Romania, health promotion

## Abstract

Background/Objectives: Parental weight stigma and bias can shape how parents talk about weight and health with their children, yet their interplay in Romania is unexplored. We examined how parents’ experienced stigma, internalized bias, and explicit antifat attitudes relate to weight- and health-focused conversations with 5–17-year-olds, and whether these links vary by child weight status. Methods: In a cross-sectional survey of 414 Romanian parents, we assessed stigma (teasing/unfair treatment), internalized bias (WBIS-M), antifat attitudes (AFA, UMBFAT), and frequency of health (healthy eating/PA) versus weight-focused talks and comments. BMI-derived child weight status was classified via WHO percentiles. Multivariate regressions and mediation analyses tested predictors and indirect effects. Results: Nearly 80% of parents discussed weight at least sometimes; higher child BMI percentile and parental internalized bias independently predicted more weight conversations (β = 0.44 and β = 0.25, both *p* < 0.001). Internalized bias mediated the effect of experienced stigma on weight talk (indirect effect = 0.105, 95% CI [0.047, 0.172]). Explicit antifat attitudes drove comments about others’ weight (β = 0.17, *p* = 0.002). Health-focused talks were unrelated to stigma or bias but were more frequent among parents with higher education, better self-rated health, and lower BMI. Conclusions: Parents’ internalized weight bias—shaped by stigma—fuels weight-focused conversations, especially when children have higher BMI, while antifat attitudes underlie negative comments about others. Interventions should reduce parental internalized bias and train supportive, health-centered communication to curb weight stigma transmission.

## 1. Introduction

According to the Global Obesity Observatory data for Romania, the national prevalence among children and adolescents aged 5–19 shows that 20.2% are overweight and 20.2% are obese. This situation is more common in rural areas (41.8%) compared to urban areas (39.2%), and among boys (45.5%) compared to girls (35.6%) [1,2]. Parental influence is a cornerstone of child development, playing a pivotal role in shaping children’s health behaviors and long-term well-being; yet how Romanian parents discuss weight within the family setting remains mainly unexamined [3]. Parents have a greater influence on children’s eating, body image, and weight than peers or media. Positive parental practices like role-modeling healthy meals predict better eating habits and body image [3,4,5]. Likewise, structured family meals foster supportive dynamics that help prevent disordered eating [6,7]. In contrast, while impactful, media portrayals of idealized bodies and peer feedback typically play a secondary role compared with the foundational attitudes and behaviors modeled by parents [8,9]. Integrating strong parental involvement with an awareness of external influences is essential for promoting children’s long-term health trajectories.

Emerging evidence highlights that parental weight bias, in its internalized, explicit, and experienced forms, profoundly shapes parental practices and child outcomes. Parents who internalize negative societal stereotypes about their own weight often respond with coercive feeding practices, such as excessive dietary restrictions, which paradoxically can foster unhealthy eating behaviors [10,11]. Similarly, parents’ explicit antifat attitudes and self-referential dieting talk can translate into derogatory comments about a child’s appearance, undermining body satisfaction and increasing the risk of disordered eating [12,13]. The stress associated with experiencing weight stigma can also disrupt consistent, supportive parenting practices [14,15].

Understanding how weight bias impacts parental mental health is crucial, as it can offer insights into how healthcare providers must commit to better supporting families in cultivating healthier eating habits and physical activity routines [16,17]. Research distinguishes between two main communication styles. Specific weight-focused communication, such as comments on a child’s weight or parental dieting talk, has been linked to adverse outcomes, including lower self-esteem and a heightened risk of disordered eating [18,19]. This type of communication often triggers feelings of shame and body dissatisfaction, which can further contribute to unhealthy eating patterns and emotional distress [20]. In contrast, general health-focused communication that emphasizes nutritious eating and physical activity tends to foster positive self-regard and healthier habits [21,22,23,24].

Despite the acknowledged importance of these factors, research has often examined parental bias and communication patterns in isolation. Less is known about the intricate interplay between parents’ experiences with stigma, their internalized biases, their explicit attitudes, and the specific types of weight-related conversations they have with their children [25,26,27,28]. Existing evidence shows that when parents internalize stigma, they often make critical comments about their child’s weight or share their dieting behaviors, which have been reported as practices linked to poorer self-esteem and body satisfaction in children [25]. Parents’ body dissatisfaction further predicts more controlling feeding practices, perpetuating stigma and unhealthy patterns [26]. In contrast, weight-neutral, health-focused discussions, emphasizing nutrition education and active play, are associated with healthier eating habits and improved well-being [29,30].

Furthermore, much of the existing research originates from North American or Western European contexts. There is a notable paucity of research investigating these sensitive psychosocial dynamics within Eastern European populations, such as Romania, where cultural factors and traditional beauty ideals may uniquely moderate communication strategies [31,32,33,34]. Understanding these dynamics in a Romanian context is a critical and unaddressed research gap.

Given the potential impact of parental factors on child health and the limited research exploring these dynamics in Romania, this study aimed to investigate the relationships between parental experienced weight stigma, internalized weight bias, explicit antifat attitudes, and various forms of parent–child communication about weight and health. The specific objectives were to: (1) Describe the prevalence of different types of parental weight-related communication and the levels of experienced stigma, internalized bias, and antifat attitudes; (2) Compare the frequency of parental communication across different child weight status categories; (3) Examine the bivariate correlations between these parental factors, communication patterns, and parent/child characteristics; (4) Identify the unique contributions of stigma, bias, and attitudes in predicting weight-related communication and (5) Test whether internalized weight bias and antifat attitudes mediate the relationship between parents’ experienced weight stigma and their communication patterns.

## 2. Materials and Methods

### 2.1. Study Design and Participants

This study utilized a cross-sectional survey design to examine parental weight bias, attitudes, and communication patterns. An a priori sample size calculation was performed prior to recruitment. Based on the total population of school-aged children in Cluj County (*N* = 176,858), it was determined that a minimum of 384 participants were required to achieve a 95% confidence level with a 5% margin of error. Our final analytic sample of *N* = 414 meets this requirement. Data were collected online via Google Forms between September and December 2024. A non-probability convenience sampling method was employed. Participants were recruited from Cluj-Napoca, Romania, using a combination of online advertisements (e.g., social media) and community sampling through partnerships with local schools participating in the broader research project.

To be eligible, participants had to be parents or primary caregivers of at least one child aged 5 to 17 years, be at least 18 years old themselves, and reside in Romania. Exclusion criteria included parents whose reported child was outside the specified age range.

A participant flow diagram is provided in Figure 1. A total of 448 individuals responded to the survey. Participants were excluded if their child did not meet the age criteria (*n* = 12), if they had substantially incomplete anthropometric data for themselves or their child (*n* = 20), or if they entered biologically implausible values for anthropometric data (*n* = 2). After these exclusions, the final analytic sample consisted of *N* = 414 parents.

All study procedures were approved by the Scientific Council of Babeș-Bolyai University Cluj-Napoca (Approval No. 224, 27 February 2024). Participants provided informed consent electronically before beginning the online survey. Participation was voluntary, anonymous, and no compensation was provided.

### 2.2. Data and Measurement

Participants completed an online survey comprising self-report measures assessing demographic characteristics, anthropometrics for both parent and child, parental communication regarding weight and health, experienced weight stigma, internalized weight bias, antifat attitudes, and fat phobia. All multi-item scales were selected based on their established use and validated psychometric properties in prior research. Cronbach’s alpha (α) coefficients reported below reflect the internal consistency reliability of multi-item scales in the current sample (*N* = 414).

#### 2.2.1. Demographics

Parents provided demographic information, including their age group (in 5-year increments), gender, and highest level of educational attainment, marital status, current employment status, and number of children. To assess health, parents rated their general health on a scale from ‘Poor’ to ‘Very good’ and indicated the presence of any long-term morbidity (Yes/No). For clarity of presentation, some variables were grouped in Table 1. For the purpose of regression and mediation analyses, several of these demographic variables were further dichotomized, as detailed in Appendix A.

#### 2.2.2. Anthropometrics

Parents’ and children’s height and weight were self-reported. Parent Body Mass Index (BMI) was calculated and categorized using standard World Health Organization (WHO) classifications (Underweight, Normal weight, Overweight, Obese). Child BMI was calculated similarly, and weight status was determined using age- and sex-specific BMI percentiles according to WHO Child Growth Standards [35]. Children were classified into four categories: Underweight (≤5th percentile), Healthy Weight (>5th to <85th percentile), Overweight (≥85th to <95th percentile), and Obese (≥95th percentile). For the purpose of regression and mediation analyses, parent BMI status, child weight status, and child gender were dichotomized, as detailed in Appendix A. Descriptive statistics for all child characteristics are presented in Table 2.

#### 2.2.3. Parental Health and Weight Conversations

Frequency of conversations was measured using a six-item scale adapted from previous research [27,36,37]. Two items assessed health conversations, asking parents how often in the past year they had talked with their child about (a) healthy eating habits and (b) being physically active. Four items assessed weight conversations, asking how often parents had talked with their child about his/her weight or size, told their child that he/she weighs too much, suggested the child should eat differently for weight reasons, or recommended exercise for weight management. Participants responded using a five-point Likert scale (1 = “Never or rarely” to 5 = “Almost every day”). An average Health Conversations score (α = 0.77) was computed using the mean of the first two items. An average Weight Conversations score (α = 0.85) was calculated using the mean of the latter four items. Higher scores indicate greater frequency.

#### 2.2.4. Parental Weight Comments

The frequency of specific parental comments made in the child’s presence was assessed using three items, adapted from previous research on parental weight talk [37,38]. These items measured comments about (a) the parent’s own weight, (b) other people’s weight, and (c) the parent’s own diet or exercise routines, each rated on a 4-point Likert scale (1 = “Never” to 4 = “Often/Very often”).

#### 2.2.5. Experienced Weight Stigma

Experienced weight stigma was measured with two yes/no questions assessing lifetime experiences of weight-based teasing or unfair treatment [37,39]. An affirmative response to either question classified the parent as having a history of experienced weight stigma.

#### 2.2.6. Internalized Weight Bias

The 10-item modified Weight Bias Internalization Scale (WBIS-M) assessed self-devaluation due to weight (α = 0.94) [40,41]. Participants responded on a 7-point Likert scale (1 = “Strongly disagree” to 7 = “Strongly agree”), and a total score was computed as the mean of all items (α = 0.94).

#### 2.2.7. Antifat Attitudes (AFA)

The 13-item Antifat Attitudes Questionnaire (AFA) measured explicit negative attitudes on a 10-point scale (0 = “Very strongly disagree” to 9 = “Extremely agree”) [42,43]. An overall mean score was computed for the total scale (α = 0.87) as well as its three established subscales: Dislike (α = 0.82), Fear of Fat (α = 0.85), and Willpower (α = 0.88).

#### 2.2.8. Universal Measure of Bias (UMBFAT)

The 20-item Fat Phobia Scale, also referred to as the Universal Measure of Bias towards Fat People (UMBFAT), was used to assess attitudes on a 7-point scale (1 = “Strongly agree” to 7 = “Strongly disagree”) [44]. After reverse-coding relevant items for consistent scoring, a total score was computed as the sum of all 20 items. Four subscale scores were also computed: Negative Judgment (α = 0.94), Distance, Attraction (α = 0.71), and Equal Rights (α = 0.96). Reliability for the total score (α = 0.84) and most subscales was acceptable to excellent, though the Distance subscale showed low reliability in this sample (α = 0.57).

### 2.3. Statistical Analysis

All statistical analyses were conducted using IBM SPSS Statistics for Mac OS (Version 29.0.1), with statistical significance set at an alpha level of 0.05. To address the study’s initial objectives, preliminary analyses included descriptive statistics to characterize the sample and key variables (Objective 1) and reliability tests (Cronbach’s alpha) for all multi-item scales.

To compare communication patterns across child weight categories (Objective 2), one-way analyses of variance (ANOVAs) were conducted. Significant main effects were followed by post hoc tests to identify specific group differences. Where the assumption of equal variances was violated, as indicated by Levene’s test, the Games-Howell test was used, as it does not require homogeneity of variances. In line with Objective 3, bivariate relationships were examined using Spearman’s rank-order correlations (ρ) due to the non-normal distribution of several variables.

To address Objective 4, which was to identify the unique predictors of parental communication, a series of hierarchical multiple linear regressions was conducted for each of the five communication outcomes. This approach was chosen to assess the predictive contribution of the main psychological factors (experienced stigma, internalized bias, and antifat attitudes) after controlling for key sociodemographic and weight-related covariates.

To test the mediation hypothesis (Objective 5), analyses were performed using the PROCESS macro for SPSS (Version 4.3) [45]. These models investigated whether internalized weight bias and antifat attitudes mediated the relationship between experienced weight stigma and the primary weight-communication outcomes, while controlling for covariates. Significance of indirect effects was determined using bootstrapped confidence intervals.

## 3. Results

### 3.1. Participant Characteristics

The final sample consisted of 414 parents residing in Romania. As detailed in Table 1, the participants were predominantly female (91.8%), married (89.6%), and had attained higher education (78.5%). The majority of parents were aged 40 years or older (58.0%). Regarding health, 77.5% of participants reported their own health as ‘Good’ or ‘Very good’, while 22.0% reported a long-term morbidity. The mean parent BMI was 24.68 (SD = 5.25), with 36.8% of the sample classified as having overweight or obesity. Detailed participant demographic characteristics are presented in Table 1.

### 3.2. Child Characteristics

Child characteristics are summarized in Table 2. Gender information was provided for 405 children; of these, slightly over half were female (55.6%, *n* = 225) and the remainder were male (44.4%, *n* = 180); 9 parents (2.2% of the total sample) selected “Prefer not to declare”. The children’s ages ranged from 5 to 17 years (M = 10.05, SD = 1.71), with the largest concentrations aged 9–10 (46.6%) and 11–12 years (38.4%). The mean child BMI was 18.09 (SD = 4.56), though this variable showed considerable positive skewness (3.64). The mean child BMI percentile was 59.29 (SD = 33.48), with a median of 75.00. Based on standard weight status classifications derived from these percentiles, 10.4% of children were classified as underweight (<5th percentile), 53.8% as healthy weight (5th to <85th percentile), 13.6% as overweight (85th to <95th percentile), and 22.2% as having obesity (≥95th percentile).

### 3.3. Parental Communication, Stigma, Bias, and Attitude Measures

Descriptive statistics for all key study measures are presented in Table 3. Overall, general health communication was the most prevalent type of interaction, with most parents reporting frequent conversations about healthy eating and physical activity. In contrast, directive weight-focused talk, such as telling a child they weighed too much or suggesting diet changes for weight reasons, was the least common. Parental comments about their own weight and diet routines were moderately common.

Nearly half the sample (44.7%) reported a history of experiencing weight stigma. Mean scores for the psychometric scales indicated relatively low levels of internalized weight bias (M = 1.95, SD = 1.23) and antifat attitudes (AFA Total M = 2.62, SD = 1.48). As shown in Table 3, the internal consistency for all multi-item scales was acceptable to excellent (Cronbach’s αs = 0.71 to 0.96), with the exception of the UMBFAT Distance subscale (α = 0.57). Detailed frequency tables are presented in the Appendix A.

### 3.4. Differences in Parental Communication by Child Weight Status

One-way ANOVAs, summarized in Table 4, indicate significant differences across child weight status groups (Underweight, Healthy Weight, Overweight, Obesity) for all three weight-related communication variables (all *p* ≤ 0.032). Post hoc tests (Games-Howell, accounting for unequal variances where necessary) revealed a clear pattern for weight conversations: parents of children with obesity reported the most frequent conversations, followed by parents of children with overweight, who in turn reported more conversations than parents of children with healthy weight or underweight (F(3, 401) = 55.91, *p* < 0.001). For comments about their own weight, parents of children with obesity reported making these comments significantly more often than parents of children with a healthy weight (F(3, 401) = 2.96, *p* = 0.032). Regarding comments about others’ weight, parents of children with healthy weight reported significantly fewer comments than parents of children with underweight or obesity (F(3, 401) = 4.85, *p* = 0.003). These findings highlight a strong association between higher child weight status and increased frequency of various forms of parental weight talk.

### 3.5. Bivariate Correlations Between Parental Psychological Factors, Communication Patterns, and Relevant Parent/Child Characteristics

Spearman’s rank-order correlations were computed to examine the relationships between parental psychological factors, communication patterns, and key characteristics. Table 5 displays the correlations among the primary study variables; the full matrix is available in the Appendix A. The most striking association was a large, positive correlation between the child’s BMI percentile and the frequency of parent–child conversations specifically about weight (ρ = 0.50, *p* < 0.001), indicating that child weight status is a primary correlate of this communication pattern. Parental weight status was also significant, with higher parental BMI showing a strong positive correlation with the parents’ own internalized weight bias (ρ = 0.42, *p* < 0.001) and a moderate link with their personal experiences of weight stigma (ρ = 0.23, *p* < 0.001).

Parental internalized weight bias, more so than experienced stigma, emerged as a robust correlate of both attitudes and communication. Higher internalized bias was strongly associated with greater fear of fat (ρ = 0.48, *p* < 0.001) and moderately with overall antifat attitudes (ρ = 0.36, *p* < 0.001). Furthermore, parents with higher internalized weight bias reported engaging more frequently in weight-specific conversations with their child (ρ = 0.24, *p* < 0.001) and making more comments about others’ weight (ρ = 0.11, *p* < 0.05) and their own weight (ρ = 0.11, *p* < 0.05). In contrast, parental antifat attitudes were most strongly linked to making comments about others’ weight (ρ = 0.21, *p* < 0.001).

Analysis of communication patterns revealed that different comment types were highly interrelated. For instance, commenting on one’s own weight was very strongly correlated with commenting on diet and exercise (ρ = 0.55, *p* < 0.001), and engaging in specific weight-focused conversations was moderately correlated with commenting on one’s own weight (ρ = 0.42, *p* < 0.001).

Finally, parental health indicators were also linked to these psychosocial factors. Poorer self-rated health was strongly associated with higher internalized weight bias (ρ = −0.36, *p* < 0.001). The morbidity measure showed an unexpected negative correlation with experienced stigma (ρ = −0.20, *p* < 0.001), which warrants cautious interpretation. Child’s age was also relevant, as parents of older children reported more frequent weight conversations (ρ = 0.17, *p* < 0.001).

### 3.6. Predictors of Parental Weight and Health Communication

To identify the unique predictors of parental communication, we conducted five hierarchical multiple linear regression analyses. Covariates were entered in Block 1, and the main psychological predictors (stigma, bias, attitudes) were entered in Block 2. A summary of these models is presented in Table 6, while the full statistical output for each model is available in the Appendix A.

The model predicting weight-specific conversations was by far the most robust, explaining a substantial portion of the variance (Adjusted R^2^ = 0.306). The addition of parental psychosocial factors significantly improved the model (ΔR^2^ = 0.048, *p* < 0.001). The child’s BMI percentile emerged as a strong, primary predictor (β = 0.44, *p* < 0.001). Higher parental internalized weight bias was also a strong, unique predictor of more frequent weight conversations (β = 0.25, *p* < 0.001). Interestingly, having personally experienced weight stigma had an opposing, small effect, predicting slightly less frequent conversations (β = −0.09, *p* = 0.049).

For other forms of weight-related talk, parental psychosocial factors also played a distinct role. Internalized weight bias was a modest predictor of making more comments about one’s own weight (β = 0.13, *p* = 0.036). In contrast, comments about others’ weight were not significantly predicted by internalized bias (*p* = 0.081) but were uniquely associated with higher explicit antifat attitudes (β = 0.17, *p* = 0.002) and higher fat phobia (UMBFAT scores; β = 0.11, *p* = 0.028).

Notably, the primary psychological predictors related to weight stigma and bias did not significantly explain the variance in more general health communication after accounting for covariates. The addition of these predictors did not improve the models for comments about diet/exercise (ΔR^2^ = 0.014, *p* = 0.243) or general health conversations (ΔR^2^ = 0.006, *p* = 0.663). These more general communication styles were instead primarily predicted by demographic and health characteristics. Higher parental education was a consistent predictor across these models (β = 0.20 for both, *p* < 0.001), as was parental health status for general health talk.

### 3.7. Mediation Analyses: The Role of Internalized Weight Bias and Attitudes

To test the hypothesis that the effect of experienced stigma on parental weight communication is transmitted through parental psychosocial factors, we conducted mediation analyses using Hayes’ PROCESS macro (Model 4), controlling for all demographic and anthropometric covariates. As depicted in Figure 2, the results showed that internalized weight bias was a significant mediator for two of the three outcomes. Specifically, a significant indirect effect was found from experienced weight stigma through internalized weight bias to more frequent weight conversations (Indirect Effect = 0.105, 95% CI [0.048, 0.172]) and more frequent parental comments about their own weight (Indirect Effect = 0.055, 95% CI [0.004, 0.116]). This suggests that experiencing stigma is associated with greater internalization of weight bias, which in turn predicts more communication about weight. In contrast, a parallel mediation model predicting comments about others’ weight found no significant indirect effects via internalized bias, antifat attitudes, or fat phobia.

## 4. Discussion

### 4.1. Principal Findings and Interpretations

This study, the first of its kind in Romania, delved into the intricate dynamics of parental experiences with weight stigma, their internalized weight bias, and their antifat attitudes. We examined how these factors influence parent–child communication about weight and health. Our findings revealed that while general health discussions are prevalent, specific weight talk is strongly linked to the child’s weight status.

This suggests such conversations are often a reaction to parental concern about a child’s body size. Notably, internalized weight bias emerged as a key psychological factor. It was an independent predictor of several communication types and acted as a crucial mechanism linking a parent’s experienced stigma to more frequent weight conversations and comments about their own weight. This finding is particularly concerning, as studies consistently show that parental self-deprecating weight talk is associated with poorer adolescent health outcomes, including disordered eating and psychological distress [46,47]. In contrast, we found that explicit antifat attitudes most strongly predicted comments about others’ weight. This mirrors evidence that parental self-referential dieting talk often translates into derogatory comments about a child’s appearance, undermining body satisfaction [12,15]. An interaction shown elsewhere to worsen adolescent well-being when directed at children themselves [47].

Our Romanian sample also reported that 79.7% of parents discuss their child’s weight at least occasionally, and 42.9% suggest dietary changes for weight control, figures lower than general health talk but still substantial. These patterns are not unique to Romania. Similar trends appear internationally: in Australia, 78% of maternal weight-and-shape comments were positive, with negative remarks rare and modulated by parent–child gender [48]; in the U.S., even infrequent negative weight comments predict adolescent depression and unhealthy control behaviors, whereas positive comments foster body appreciation [47]. Qualitative work from Sweden shows parents welcome early, supportive weight dialogue but resist judgmental framing [49], and a review of ethnic-minority families highlights how cultural norms equating “chubbiness” with good parenting can discourage explicit weight-control messages [50]. These findings underscore the global relevance of our study and the powerful influence of cultural context on parent–child conversations about weight and health.

### 4.2. The Nuanced Role of Stigma, Bias, and Health Communication

Interestingly, our analysis uncovered a more complex role for experienced stigma. When internalized bias was accounted for, the link between experienced stigma and weight conversations became slightly negative. This counterintuitive finding may indicate a protective mechanism. Parents who have faced stigma but have not internalized it may consciously avoid weight talk to shield their children from the harm they themselves have experienced [46,51]. This underscores non-internalization as a potential point of resilience. A probable explanation is social desirability bias, where parents understate antifat attitudes to appear supportive, a pattern that contrasts with implicit pro-thin preferences detected elsewhere [18]. Measurement format also matters: stigma was captured via a binary yes/no item, whereas WBIS-M and AFA aggregate multiple Likert items, potentially diluting moderate agreement into low means [52]. Furthermore, protective reframing may lower self-reports, as many parents attribute teasing to societal, not personal, failings [47,53]. Moreover, the chronic stress associated with weight stigma can disrupt consistent parenting practices, leading to fluctuating feeding approaches and communication styles [14,15].

In contrast to the psychologically driven nature of weight talk, our results showed that general health conversations were not significantly predicted by parental stigma or bias. Instead, more frequent discussions about healthy eating and physical activity were most consistently associated with higher parental education. Frequent health-focused conversations have been associated with improved dietary quality, increased physical activity, and enhanced psychosocial well-being in children [21,22]. This suggests general health promotion may be a more universally accepted parenting practice, particularly among those with more educational resources. Importantly, this type of supportive, health-focused communication is consistently linked to healthier food choices and improved dietary patterns in children, positioning it as a clear avenue for positive intervention [5,54].

One of our most striking findings is the robust link between higher child weight status and more frequent parental weight-related conversations. This strong association, confirmed in both our correlational and regression analyses, shows that parents of children with higher BMI are significantly more likely to initiate these discussions [25,47]. However, the content of these conversations matters deeply: negative remarks correlate with disordered eating, depression, and greater internalization of weight bias in adolescents [18,47], highlighting the potential harm of such comments. In contrast, positive comments support body appreciation and lower bias internalization [47,55]. Similarly, positive parental dietary practices, such as meal modeling and encouragement of nutritious eating, have also been linked to reduced risk of disordered eating and improved body image in children [3,4,5]. This supportive dialogue aligns with evidence that structured family meals foster healthy eating behaviors and positive body image in children [6,7].

Our mediation analyses reveal that internalized weight bias is a crucial mechanism that links a parent’s experienced stigma to their own weight talk. This finding underscores the significant role of societal stigma in shaping family conversations. Parents who internalize stigma not only talk about weight more often but do so through a lens of self-criticism, potentially undermining children’s emotional well-being [56,57]. This highlights the need for a broader understanding of the social context where these conversations occur. In contrast, explicit antifat attitudes were predominantly associated with comments about others’ weight. This reflects a social-judgment pathway that can perpetuate stigma within family and community contexts [58].

Interestingly, when internalized bias is accounted for, the link between experienced stigma and weight conversations becomes slightly negative. This counterintuitive finding aligns with evidence that parents who do not internalize stigma may consciously avoid weight talk to shield themselves and their children from its harmful effects [59,60]. This underscores the urgent need for interventions that reduce parents’ self-directed bias, thereby fostering more supportive, health-focused dialogues—and challenge explicit antifat beliefs to curb negative remarks about others.

### 4.3. Limitations

This study offers important insights, particularly as a first exploration of these complex dynamics within the Romanian context using validated measures and mediation analysis to uncover potential underlying mechanisms. Despite these strengths, it is crucial to acknowledge several key limitations that inform the interpretation and generalizability of our findings.

First and foremost, the cross-sectional design provides a critical snapshot of associations but precludes causal inference. Our findings establish a strong theoretical basis for how parental stigma may influence communication, but we cannot determine the long-term effects of these communication styles on child weight trajectories. Similarly, this design cannot capture the dynamic, bidirectional nature of family communication, where a child’s responses undoubtedly shape subsequent parental behaviors. These questions of causality and reciprocal influence are paramount and should be addressed in future longitudinal research that builds upon the associations identified here.

Second, our reliance on self-report measures for all psychological constructs and communication patterns introduces potential biases. These include social desirability bias, which may be particularly pronounced among highly educated parents who are more aware of socially approved parenting practices, and recall bias, where participants may not accurately remember the frequency or nature of past conversations. The use of a single method for data collection also raises concerns about shared method variance, potentially inflating observed correlations. Furthermore, our data is based solely on parent-reported outcomes, without validation from the child’s perspective or through objective behavioral observation, which limits the scope of our conclusions. Future research could benefit from integrating objective observations or reports from multiple family members. Furthermore, this study’s primary focus was on psychosocial drivers of communication, not clinical ones. Consequently, our assessment of parental and child health did not quantify specific obesity-related co-morbidities (e.g., metabolic syndrome, hypertension). While our findings on the role of internalized bias are robust across general health status, we cannot determine how the presence of specific weight-related diseases might moderate parental communication. This clinical dimension represents a crucial and complementary avenue for future investigation.

Third, the sample’s homogeneity significantly limits the external validity and generalizability of our findings. As noted, the sample was heavily unbalanced, with 91.8% of participants being female and 78.5% holding a tertiary education. This specific demographic profile, primarily composed of highly educated mothers from Cluj-Napoca, means our results may not accurately reflect the experiences or communication patterns of fathers, parents from diverse socioeconomic backgrounds, or those residing in other regions of Romania. This imbalance could have influenced the results, and therefore, broader sampling strategies are essential for future studies to ensure more representative findings.

Fourth, a methodological limitation stems from our sample size calculation. As this was an initial, exploratory study in this context, the sample size was determined with the primary goal of achieving demographic representativeness for the target population rather than being based on an a priori power analysis for a specific outcome. While this is a limitation, the achieved sample of *N* = 414 provided sufficient statistical power to robustly detect the moderate-to-large associations that were central to our research questions and to generate clear hypotheses for future confirmatory studies.

Fifth, the use of self-reported anthropometric data for both parents and children is a notable limitation. While practical for survey-based research, self-reported height and weight are known to be prone to systematic inaccuracies. There is a well-documented tendency for individuals to underestimate their own weight or that of their children, particularly in categories of overweight or obesity. This could have introduced measurement error and potentially attenuated or exaggerated associations with other variables. Wherever feasible, future research should incorporate objectively measured anthropometric data.

Sixth, the observed low internal consistency reliability (Cronbach’s α = 0.57) of the UMBFAT Distance subscale warrants cautious interpretation of any findings related to this specific construct. While other measures demonstrated good to excellent reliability, this particular subscale’s performance suggests it may not have consistently captured the intended concept in our sample.

Seventh, the interpretation of our statistical results requires caution on two fronts. Given the number of statistical tests performed to explore various associations, there is an increased possibility of Type I errors, or false positives. Additionally, several statistically significant findings, particularly in models predicting comments about others’ weight or diet, were associated with very small effect sizes. It is crucial to distinguish statistical significance from practical or clinical relevance, and these small effects suggest that while the relationships may be real, their impact may be limited in a real-world context.

Finally, while our measures captured the frequency of different communication types, they did not delve into the qualitative aspects of these interactions. We lack information on the tone, specific content, or emotional context of these conversations. For instance, a “weight conversation” could be delivered in a shaming or a supportive manner, with vastly different implications for a child’s well-being. This nuance is critical for developing effective interventions.

### 4.4. Implications for Practice

Our study underscores the urgent need for interventions that directly address parents’ internalized weight bias to prevent self-devaluation and the subsequent use of communication patterns linked to problematic eating behaviors, body dissatisfaction, and the intergenerational transmission of weight stigma [47,48]. Programs like the Cognitive-Behavioral Weight Bias Internalization and Stigma (BIAS) intervention and Confidence Body, Confident Child (CBCC) are evidence-based models that have shown promise in reducing internalized bias and equipping parents with strategies for supportive dialogue [61,62,63]. Healthcare professionals, including pediatricians, family physicians, and registered dietitians, are uniquely positioned to leverage these findings. They can incorporate routine screening for parental weight bias during consultations and integrate stigma-reduction techniques into family-centered care [17,21]. Beyond clinical settings, schools and public health initiatives can play a vital role by providing educational resources to parents and promoting weight-inclusive health curricula.

Best practices should prioritize framing conversations around health and body functionality rather than appearance or weight. Adopting principles from frameworks such as Health at Every Size (HAES) can guide this shift, helping to reduce weight stigma and support holistic well-being [64]. Encouraging intuitive eating principles and normalizing diverse body types are vital strategies to foster positive body image and interrupt intergenerational weight stigma [65,66]. These health-focused discussions also empower children to internalize positive body-image messages and adopt healthier behaviors autonomously, building resilience against external societal stigma [23,24,67,68]. Finally, these findings reinforce the urgent need for educational interventions that train parents in effective, health-centered communication strategies [69,70].

### 4.5. Future Research Directions

While this study provides important initial insights, its limitations underscore several critical avenues for future research to foster a more comprehensive and actionable understanding of family weight communication. Methodologically, future investigations should transition from cross-sectional designs to longitudinal approaches to establish clearer cause-and-effect relationships between parental psychological factors, communication patterns, and children’s long-term behavioral and psychosocial outcomes. To capture the essential nuance currently missing, these studies should integrate qualitative methods (e.g., in-depth interviews, focus groups, observational studies) to explore the tone, specific content, emotional context, and perceived impact of weight and health discussions from both parent and child perspectives. Enhancing data accuracy and richness is vital through the incorporation of objective anthropometric measures where feasible, and by collecting dyadic data directly from both parents and children, which would allow for examining discrepancies in perceptions within the family unit. Finally, there is a critical need for rigorous intervention trials to test the efficacy, long-term impact, and scalability of programs designed to reduce parental internalized bias and improve family communication about health.

Beyond methodological rigor, expanding the scope of inquiry is crucial. Future research must prioritize recruiting more diverse and representative samples across various socioeconomic strata, educational backgrounds, and geographic locations, and ensure a balanced representation of fathers to significantly enhance the external validity and generalizability of findings. Given that cultural norms profoundly shape parental feeding practices and weight communication, future work must examine these dynamics in diverse cultural contexts to develop culturally sensitive and relevant interventions [71,72]. Concurrently, exploring additional mediating pathways (e.g., parental mental health, broader family environment) and moderating factors (e.g., child’s developmental stage, cultural background, family support systems) will further elucidate the complex interplay between parental bias, communication strategies, and their multifaceted influence on child health and well-being, ultimately informing more targeted and culturally sensitive interventions.

## 5. Conclusions

This study, representing an initial exploration in the Romanian context, suggests that parents’ internalized weight bias, which appears to be influenced by prior experiences of stigma, is significantly associated with the frequency of weight-focused conversations with their children, especially when children have higher BMI percentiles. In contrast, explicit antifat attitudes and fat phobia seem to be more strongly linked to parental comments about others’ weight, indicating a distinct social-judgment pathway. Our findings also suggest that general health-focused discussions are largely unaffected by parental bias or stigma, instead showing correlations with higher parental education, better self-rated health, and lower parental BMI.

These associations suggest that addressing parental internalized bias could be an important component of future interventions. Efforts aimed at reducing self-directed stigma and encouraging a reframing of weight talk around health and body functionality might help foster more supportive family dialogues. Therefore, educational initiatives, potentially integrated into pediatric and nutrition care, may represent a valuable opportunity to equip parents with skills to engage in empowering, health-centered communication.

## Figures and Tables

**Figure 1 nutrients-17-02920-f001:**
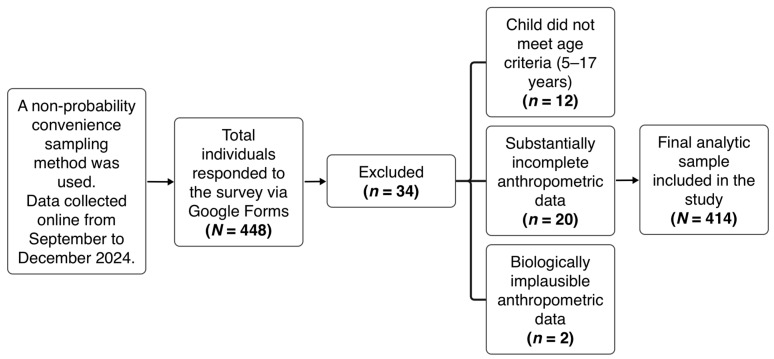
Participant Flow Diagram.

**Figure 2 nutrients-17-02920-f002:**
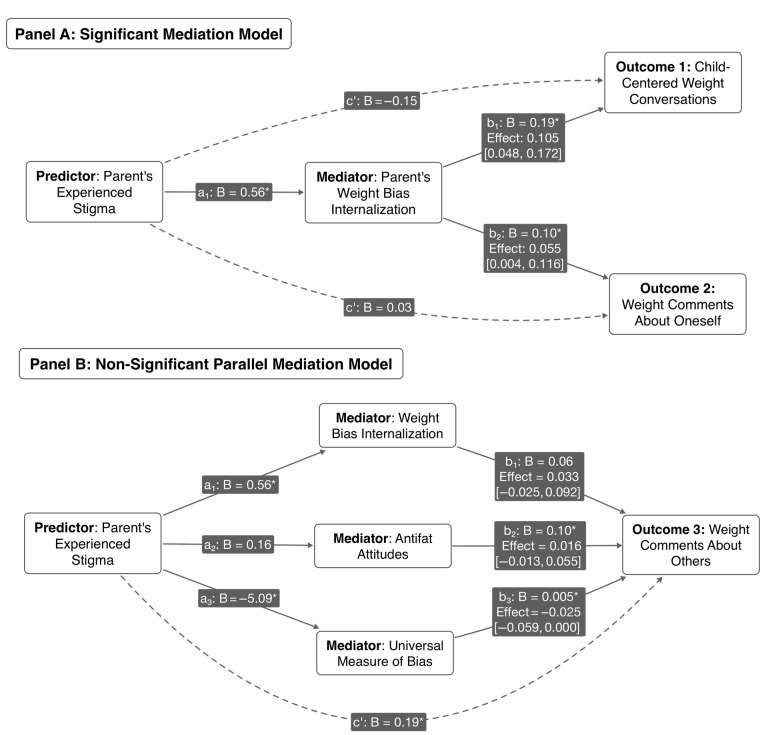
The Mediating Role of Parental Psychosocial Factors in Parental Weight Communication. Note. Panel (**A**) shows that internalized weight bias significantly mediated the relationship between experienced weight stigma and two communication outcomes. Panel (**B**) shows the non-significant parallel mediation model for comments about others’ weight. All models controlled for parent age group, gender, education, marital status, self-rated health, parent BMI, child age, gender, child BMI percentile, and number of children. The 95% confidence intervals for the indirect effects were derived from 5000 bootstrap samples. An asterisk (*) denotes a path coefficient is significant at *p* < 0.05. Dashed lines represent direct pathways, whereas solid lines represent pathways through mediators.

**Table 1 nutrients-17-02920-t001:** Participant Demographic and Background Characteristics (*N* = 414).

Characteristic	Category	*n*	%
Age Group	<40 years	174	42.0
	≥40 years	240	58
Gender	Female	380	91.8
	Male	32	7.7
	Prefer not to declare	2	0.5
Educational Attainment	Lower Education	89	21.5
	Higher Education	325	78.5
Marital Status	Married	371	89.6
	Divorced/Widowed/Never married	43	10.4
Employment Status	Employed	291	70.3
	Domestic worker ^a^	38	9.2
	Entrepreneur/Self-employed	85	20.5
Number of Children ^b^	1 Child	149	37.1
	2 Children	216	53.7
	3 Children	37	9.2
BMI Classification	Underweight (<18.5)	11	2.7
	Normal weight (18.5–24.9)	251	60.6
	Overweight (25–29.9)	100	24.2
	Obese (≥30)	52	12.6
Self-Rated Health	Poor	2	0.5
	Acceptable	79	19.1
	Good	208	50.2
	Very good	113	27.3
	Don’t know/Prefer not to answer	12	2.9
Long-term Morbidity	Yes	91	22
	No	273	65.9
	Don’t know/Prefer not to answer	50	12.1

Note. Percentages are calculated based on the total sample (*N* = 414) unless otherwise specified. ^a^ This category represents participants who identify primarily as homemakers or perform domestic work within their household. ^b^ Valid *N* = 402; percentages are based on valid cases. 12 participants (2.9% of the total) had missing data for this variable.

**Table 2 nutrients-17-02920-t002:** Child Characteristics.

Characteristic	Statistic	*n*	Value ^a^
Gender	Female	225	54.30%
	Male	180	43.50%
	Prefer not to declare	9	2.20%
Age (Years)	Mean (SD)	414	10.05 (1.71)
	Range	414	5–17
BMI	Mean (SD)	414	18.09 (4.56)
	Range	414	12.08–62.22
BMI Percentile ^b^	Mean (SD)	405	59.29 (33.48)
	Median	405	75
	Range	405	5.00–100.00
Weight Status ^b^	<5th (Underweight)	42	10.40%
	5th–<85th (Healthy Weight)	218	53.80%
	85th–<95th (Overweight)	55	13.60%
	≥95th (Obesity)	90	22.20%

Note. *N* = 414 unless otherwise specified. SD = Standard Deviation; BMI = Body Mass Index. Short variable names used in analyses are shown in parentheses. ^a^ Percentages for categorical variables are calculated based on the valid number of cases for that variable. ^b^ Based on *N* = 405; excludes 9 participants for whom child gender was not declared, preventing calculation of BMI percentile and weight status. Weight status categories are defined using standard BMI percentile cutoffs.

**Table 3 nutrients-17-02920-t003:** Descriptive Statistics, Range, and Reliability for Key Study Measures (*N* = 414).

Measure	M	SD	Possible Range	Observed Range	Cronbach’s α
**Communication**					
Health Conversations	3.84	0.73	1–5	1.00–5.00	0.77
Weight Conversations	2.31	0.96	1–5	1.00–5.00	0.85
Comments Own Weight	2.74	0.89	1–4	1–4	N/A
Comments Others’ Weight	2.01	0.88	1–4	1–4	N/A
Comments Diet/Exercise	2.76	0.87	1–4	1–4	N/A
**Stigma & Bias/Attitudes**					
Experienced Stigma (% Yes) ^a^	44.70%	-	0–1	0–1	N/A
Internalized Weight Bias	1.95	1.23	1–7	1.00–7.00	0.94
AFA Total	2.62	1.48	0–9	0.00–8.46	0.87
AFA Dislike	1.87	1.39	0–9	0.00–8.00	0.82
AFA Fear	3	2.31	0–9	0.00–9.00	0.85
AFA Willpower	3.99	2.42	0–9	0.00–9.00	0.88
UMBFAT Total ^b^	65.99	20.16	20–140	22–129	0.84
UMBFAT Negative Judgment ^b^	14.25	9.13	5–35	5–35	0.94
UMBFAT Distance ^b^	16.33	6	5–35	5–35	0.57
UMBFAT Attraction ^b^	19.85	6.05	5–35	5–35	0.71
UMBFAT Equal Rights ^b^	15.56	10.23	5–35	5–35	0.96

Note. M = Mean; SD = Standard Deviation; AFA = Antifat Attitudes Questionnaire; UMBFAT = Universal Measure of Bias (Fat Phobia Scale); N/A = Not Applicable. Reliability (Cronbach’s α) reported for multi-item scales. Bold text refers to the main study measures categories. ^a^ Percentage reporting ‘Yes’ to experiencing teasing or unfair treatment due to weight. ^b^ UMBFAT scores are sums of items (each item 1–7); Possible Range reflects min/max possible sum.

**Table 4 nutrients-17-02920-t004:** Differences in parents’ weight-related communication by child weight status (*N* = 405).

Variable/Group	*N*	M (SD)	Test Statistic	*df*	*p*-Value	Games-Howell
Weight Conversations			F = 55.91	3, 401	<0.001	UW = HW < OW < OB
Underweight	42	1.87 (0.69) ^a^				
Healthy Weight	218	1.96 (0.73) ^a^				
Overweight	55	2.51 (0.92) ^b^				
Obesity	90	3.21 (0.96) ^c^				
Comments about Own Weight			F = 2.96	3, 401	0.032	HW < OB
Underweight	42	2.74 (0.89) ^ab^				
Healthy Weight	218	2.64 (0.89) ^a^				
Overweight	55	2.75 (0.91) ^ab^				
Obesity	90	2.97 (0.85) ^b^				
Comments about Others’ Weight			F = 4.85	3, 401	0.003	HW < OB
Underweight	42	2.24 (0.93) ^ab^				
Healthy Weight	218	1.86 (0.79) ^a^				
Overweight	55	2.11 (0.96) ^ab^				
Obesity	90	2.20 (0.96) ^b^				

Note. M = Mean; SD = Standard Deviation. Superscripts (^a–c^) within rows indicate groups that differ significantly based on Games-Howell post hoc tests (*p* < 0.05); groups sharing a superscript letter are not significantly different. Levene’s test for homogeneity of variances was significant (*p* < 0.001) for Weight Conversations, indicating unequal variances; therefore, Games-Howell post hoc results are interpreted. Levene’s test was not significant for other variables, but Games-Howell results are presented for consistency. UW = Underweight, HW = Healthy Weight, OW = Overweight, OB = Obesity.

**Table 5 nutrients-17-02920-t005:** Spearman Correlations Among Key Parental Psychosocial Factors and Outcome Variables.

Variable	Experienced Stigma	Internalized Weight Bias	AFA Total	UMBFAT Total
Communication Patterns				
Health Conversations	−0.01	−0.08	0.06	−0.05
Weight Conversations	0.01	0.24 ***	0.04	0.09
Comments: Own Weight	0.06	0.11 *	0.12 *	0.03
Comments: Others’ Weight	0.12 *	0.11 *	0.21 ***	0.14 **
Comments: Diet/PA	0.02	−0.04	0.12 *	−0.01
Parent/Child Characteristics				
Parental SRH	−0.14 **	−0.36 ***	−0.14 **	−0.03
Parental Chronic Morbidity	−0.20 ***	−0.17 **	−0.07	−0.02
Parental BMI	0.23 ***	0.42 ***	0.05	0.01
Child’s Age	−0.09	−0.08	−0.12 *	−0.02
Child BMI Percentile	0.12 *	0.21 ***	−0.02	0.02

Note. Coefficients are Spearman’s rho (*r*s). AFA = Antifat Attitudes Questionnaire; UMBFAT = Universal Measure of Bias (Fat Phobia Scale); PA = Physical Activity; SRH = Self-Rated Health; BMI = Body Mass Index. N varies slightly for correlations involving SRH (*n* = 402), Morbidity (*n* = 364), and Child BMI Percentile (*n* = 405) due to missing data. * *p* < 0.05. ** *p* < 0.01. *** *p* < 0.001.

**Table 6 nutrients-17-02920-t006:** Summary of Hierarchical Regression Analyses Predicting Parental Communication Styles.

Dependent Variable	Overall Model Adj. R^2^	ΔR^2^ for Main Predictors	Significant Predictors (β)
Weight Conversations	0.306 ***	0.048 ***	Covariates: Child Age (0.18 ***), Child BMI percentile (0.44 ***) Main Predictors: Experienced Stigma (−0.09 *), Internalized Bias (0.25 ***)
Comments: Own Weight	0.092 ***	0.023 *	Covariates: Parent Education (0.20 ***), Child Age (0.21 ***) Main Predictors: Internalized Bias (0.13 *)
Comments: Others’ Weight	0.084 ***	0.082 ***	Covariates: Parent BMI (−0.12 *) Main Predictors: Antifat Attitudes (0.17 **), UMBFAT (0.11 *)
Comments: Diet/Exercise	0.039 **	0.014	Covariates: Parent Education (0.20 ***) Main Predictors: None
Health Conversations	0.045 **	0.006	Covariates: Parent Education (0.20 ***), Parent SRH (0.15 **), Parent BMI (−0.12 *) Main Predictors: None

Note. *N* = 384. Full model statistics are available in Appendix A. β = standardized regression coefficient. ΔR^2^ reflects the change in variance explained by adding the main predictors (stigma, bias, attitudes) in Block 2. SRH = Self-Rated Health; BMI = Body Mass Index; UMBFAT = Universal Measure of Bias (Fat Phobia Scale). * *p* < 0.05. ** *p* < 0.01. *** *p* < 0.001.

## Data Availability

The data supporting this study’s findings are available from Babeș-Bolyai University. Access to the data can be requested from the corresponding author upon reasonable request with the permission of Babeș-Bolyai University. They are not publicly available due to privacy and ethical restrictions.

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
