# Peer review of "Talking About Weight with Children: Associations with Parental Stigma, Bias, Attitudes, and Child Weight Status"

_nutrients, 2025, doi:10.3390/nu17182920_

Round 1

Reviewer 1 Report

Comments and Suggestions for Authors

Introduction: What is the age group of Romanian boys and girls. It would be good to present the age group, so we will have better idea from the intervention point of view.

The introduction section is extremely lengthy, some of the content of methodology are also mixed with the introduction section. Omit the second paragraph of introduction.  Summarize your introduction to maximum of one page.

Methodology: Under the sub-section 2.2.1 Demography - there is no need to write the coding. If you want to present the coding then you can present all those coding in the supplementary file. Moreover, write the categories of each variable together with variable at once. No need to repeat.

It would be good to show that you calculated the BMI, we will understand that the centimetre was converted into meters. Remove the coding from section 2.2.2. Avoid overloading the main text with every coding detail and reverse-coded item.

Keep the methods narrative focused on what was measured, how, and why, not every operational nuance. Right now, the analysis plan reads like “test everything possible.”

Given the number of tests, interpret significant results cautiously and mention the possibility of false positives in the discussion.

Result: Reduce the age categories. Moreover, in the method section 2.21. it was written two categories of age, but in the result section is over 5. 

Merge various categories of education. 

Categories of self rated and long term morbidity are contradictory in the method and result section.

Need to summarize content presented in subsection 3.3. Just write what is max and min.

Mediation results are well explained but could be reduced to one short paragraph + diagram in main text.

Discussion:

Content in the discussion is repetitive of the introduction.

Need to summarize limitations, study implications and future directions.

Comments on the Quality of English Language

I found that the sentences were very lengthy. 

Author Response

Comment 1: "Introduction: What is the age group of Romanian boys and girls. It would be good to present the age group, so we will have better idea from the intervention point of view."

Response 2: Thank you for the valuable suggestion to provide more specific demographic context. Following your feedback, we have revised the introduction. The text now begins immediately with the most relevant national prevalence data for Romanian children and adolescents aged 5-19, including key demographic disparities. This provides a strong, focused rationale for the subsequent discussion on the unexamined role of parental influence in this context.

Revised text lines 40-43: "According to the Global Obesity Observatory data for Romania, the national prevalence among children and adolescents aged 5-19 shows that 20.2% are overweight and 20.2% are obese. This situation is more common in rural areas (41.8%) compared to urban areas (39.2%), and among boys (45.5%) compared to girls (35.6%) [1,2]."

Comment 2: "The introduction section is extremely lengthy, some of the content of methodology are also mixed with the introduction section. Omit the second paragraph of introduction.  Summarize your introduction to maximum of one page."

Response 2: Thank you for this valuable feedback. We agree that the original introduction was overly long and lacked a clear focus. In response, we have undertaken a comprehensive revision to shorten the section and improve its logical flow significantly. Following your suggestion, the paragraph that prematurely detailed the study's structure has been removed entirely, and the overall length has been substantially reduced to enhance readability and conciseness.

Comment 3: "Methodology: Under the sub-section 2.2.1 Demography - there is no need to write the coding. If you want to present the coding then you can present all those coding in the supplementary file. Moreover, write the categories of each variable together with variable at once. No need to repeat."

Response 3: Thank you for this advice on improving the manuscript's readability. We have removed all detailed coding for dichotomized variables from the main text to improve readability, with the suggestion that these details can be found in a supplementary file. The variable descriptions have been streamlined.

Revised text lines 330-336: "Parents provided demographic information, including their age group (in 5-year increments), gender, and highest level of educational attainment, marital status, current employment status, and number of children. To assess health, parents rated their general health on a scale from 'Poor' to 'Very good' and indicated the presence of any long-term morbidity (Yes/No). For clarity of presentation, some variables were grouped in Table 1. For the purpose of regression and mediation analyses, several of these demographic variables were further dichotomized, as detailed in Supplementary Table S1."

Comment 4: "It would be good to show that you calculated the BMI, we will understand that the centimetre was converted into meters. Remove the coding from section 2.2.2. Avoid overloading the main text with every coding detail and reverse-coded item."

Response 4: Thank you for the suggestion. We have edited the Anthropometrics section to be more concise, removing redundant statements about calculations and unit conversions. All detailed coding for dichotomized variables and reverse-coded items has also been removed from this section to improve clarity and focus.

Revised text lines 337-435: " Parents' and children's height and weight were self-reported. Parent Body Mass Index (BMI) was calculated and categorized using standard World Health Organization (WHO) classifications (Underweight, Normal weight, Overweight, Obese). Child BMI was calculated similarly, and weight status was determined using age- and sex-specific BMI percentiles according to WHO Child Growth Standards [35]. Children were classified into four categories: Underweight (≤5th percentile), Healthy Weight (>5th to <85th percentile), Overweight (≥85th to <95th percentile), and Obese (≥95th percentile). For the purpose of regression and mediation analyses, parent BMI status, child weight status, and child gender were dichotomized, as detailed in Supplementary Table S1. Descriptive statistics for all child characteristics are presented in Table 2."

Comment 5: "Keep the methods narrative focused on what was measured, how, and why, not every operational nuance. Right now, the analysis plan reads like “test everything possible.”

Response 5: We thank the reviewer for this insightful feedback. We agree that the original statistical analysis section was overly detailed and resembled an operational list rather than a strategic overview. We have substantially revised and condensed this section to focus on the rationale and purpose of our analytical approach, clarifying what was tested and why, while removing repetitive lists of variables and other operational nuances.

Comment 6: Given the number of tests, interpret significant results cautiously and mention the possibility of false positives in the discussion.

Response 6: We thank the reviewer for raising this valid methodological concern. In response, we have added a new paragraph to the limitations section that acknowledges the multiple statistical tests conducted in our study. This new text highlights the potential for an inflated risk of Type I errors (false positives) and advises that the significant results should be interpreted with appropriate caution.

Comment 7: "Result: Reduce the age categories. Moreover, in the method section 2.21. it was written two categories of age, but in the result section is over 5."

Response 7: Thank you for highlighting this inconsistency. We agree that the presentation of age categories needs to be simplified and aligned with our analytical approach. We have revised Table 1 to reflect the dichotomized age groups (<40 years and ≥40 years) that were used in our regression models, which improves clarity and consistency with the Methods section.

Comment 8: "Merge various categories of education."

Response 8: We have merged the seven granular educational attainment categories into the two main analytical groups as described in the Methods: 'Lower Education' and 'Higher Education'. The revised Table 1 now presents the data in this simplified format.

Comment 9: "Categories of self-rated and long-term morbidity are contradictory in the method and result section."

Response 9:  To resolve the inconsistency between the Methods and Results sections, we have revised the 'Self-Rated Health' categories in Table 1 to match the dichotomized groups used in our analyses (Poor/Acceptable vs. Good/Very good). The presentation for 'Long-term Morbidity' remains as Yes/No, which is consistent with the analytical approach.

Comment 10: Need to summarize content presented in subsection 3.3. Just write what is max and min.

Response 10: Thank you for the suggestion to improve the conciseness of this section. We agree that the original text was overly detailed with descriptive statistics. We have revised the paragraph to provide a high-level summary that highlights the most and least prevalent communication behaviors, as requested. The revised text now directs the reader to Table 3 for the specific means, standard deviations, and frequency data.

Revised Text lines 718-731: " Descriptive statistics for all key study measures are presented in Table 3. Overall, general health communication was the most prevalent type of interaction, with most parents reporting frequent conversations about healthy eating and physical activity. In contrast, directive weight-focused talk, such as telling a child they weighed too much or suggesting diet changes for weight reasons, was the least common. Parental comments about their own weight and diet routines were moderately common. Nearly half the sample (44.7%) reported a history of experienced weight stigma. Mean scores for the psychometric scales indicated relatively low levels of internalized weight bias (M=1.95, SD=1.23) and antifat attitudes (AFA Total M=2.62, SD=1.48). As shown in Table 3, the internal consistency for all multi-item scales was acceptable to excellent (Cronbach’s αs = .71 to .96), with the exception of the UMBFAT Distance subscale (α=.57). Detailed frequency tables are presented in the Supplementary Tables S2 to S5."

Comment 11: Mediation results are well explained but could be reduced to one short paragraph + diagram in main text.

Response 11: Thank you for this excellent suggestion to improve the communication and accessibility of our mediation findings. We agree that a more concise presentation would be more impactful. Accordingly, we have condensed the description of the mediation analyses into a single, focused paragraph and have created a diagram (Figure 2) that visually summarizes the tested models and their outcomes. This figure has been added to the manuscript, with a callout in the revised text.

Revised text lines 1000-1012: " To test the hypothesis that the effect of experienced stigma on parental weight communication is transmitted through parental psychosocial factors, we conducted mediation analyses using Hayes’ PROCESS macro (Model 4), controlling for all demographic and anthropometric covariates. As depicted in Figure 1, the results showed that internalized weight bias was a significant mediator for two of the three outcomes. Specifically, a significant indirect effect was found from experienced weight stigma through internalized weight bias to more frequent weight conversations (Indirect Effect = 0.105, 95% CI [0.048, 0.172]) and more frequent parental comments about their own weight (Indirect Effect = 0.055, 95% CI [0.004, 0.116]). This suggests that experiencing stigma is associated with greater internalization of weight bias, which in turn predicts more communication about weight. In contrast, a parallel mediation model predicting comments about others’ weight found no significant indirect effects via internalized bias, antifat attitudes, or fat phobia."

Comment 12: Content in the discussion is repetitive of the introduction.

Response 12: Thank you for this helpful observation. We carefully revised the opening of the Discussion to avoid overlap with the Introduction. Specifically, we replaced the previous opening paragraph (which restated prevalence data and detailed percentages) with a shorter, more focused paragraph summarizing only the key study aims and main findings. In addition, throughout Sections 4.1 and 4.3 we streamlined the text by removing repeated numerical results and instead highlighted the conceptual meaning of the findings supported by relevant literature. These revisions reduce redundancy and ensure that the Discussion complements rather than duplicates the Introduction.

Comment 13: Discussion: Need to summarize limitations, study implications and future directions.

Response 13: We thank the reviewer for this suggestion. We agree that a clear summary of limitations is essential for the Discussion section. In reviewing the feedback from all reviewers, we noted that several others requested a substantial expansion of this section to address critical points they felt were missing (e.g., sample biases, measurement issues, statistical interpretation). Therefore, while a simple summary of the original text would not have been sufficient, we have thoroughly revised and expanded the limitations section in a structured manner. This comprehensive section now provides the necessary foundation to effectively summarize these key points as we transition to discussing the study's implications and future research directions. Following our revision of the limitations section, we have thoroughly revised the subsequent section on "Implications for Practice and Future Research." We have organized the text under clear subheadings to distinctly present the practical implications of our findings and then outline specific, actionable directions for future studies, ensuring the discussion concludes with a clear summary of the study's contributions and next steps.

Comment 14: I found that the sentences were very lengthy. 

Response 14: To enhance readability, we have carefully edited the manuscript. We focused on simplifying sentence structures and breaking down longer, more complex sentences into shorter, clearer statements to ensure our points are communicated more effectively.

Reviewer 2 Report

Comments and Suggestions for Authors

The Authors persented very interesting and up-to-date paper about associations with parental stigma, attitudes and child weight status. This topic is extremely important considering the fact that the percentage of overweight/obese subjects among children and adolescents is high, especially in developed countries, and there is an increasing trend in the incidence in the next decades. In addition to the obvious impact of overweight/obesity on health and the risk of cardiometabolic diseases, it is important to pay attention to the factors shaping the behavior and choices of young people, including the experiences and attitudes of parents.

The study is well designed, the methodology and disscusion sections are sufficiently described. I only have two minor comments about the results which can improve their readability and understanding

1) please correct values below 1 (p-values, correlation coefficients) throughout the manuscript so that the numbers start with 0, e.g. "0.001" instead of ".001";

2) tables 5 and 6 should be shortened and include only the most important and statistically significant results; please also discuss only the most important results in the text.

Author Response

Comment 1: "The Authors persented very interesting and up-to-date paper about associations with parental stigma, attitudes and child weight status. This topic is extremely important considering the fact that the percentage of overweight/obese subjects among children and adolescents is high, especially in developed countries, and there is an increasing trend in the incidence in the next decades. In addition to the obvious impact of overweight/obesity on health and the risk of cardiometabolic diseases, it is important to pay attention to the factors shaping the behavior and choices of young people, including the experiences and attitudes of parents."

Response 1: We sincerely thank the reviewer for these encouraging comments and for highlighting the importance of our study. We fully agree that parental experiences, attitudes, and communication patterns play a critical role in shaping children’s health behaviors and weight-related outcomes, particularly in the context of rising childhood obesity worldwide. We are grateful that the reviewer recognizes the relevance and timeliness of our contribution.

Comment 2: "The study is well designed, the methodology and disscusion sections are sufficiently described. I only have two minor comments about the results which can improve their readability and understanding: 1) please correct values below 1 (p-values, correlation coefficients) throughout the manuscript so that the numbers start with 0, e.g. "0.001" instead of ".001"; 2) tables 5 and 6 should be shortened and include only the most important and statistically significant results; please also discuss only the most important results in the text."

Response 2:  Thank you for this excellent feedback aimed at improving the clarity and impact of our results. We have implemented all suggested changes. All statistical values less than 1 throughout the manuscript now include a leading zero. To improve readability, we created a new, condensed Table 5 for the main text that highlights the most important correlations. To ensure full transparency, the original, complete correlation matrix has been moved to the Supplementary Materials (Table S6). The results section has also been substantially revised to be more concise and to focus on the most statistically and theoretically significant findings, as you suggested. Additionally, we converted Table 6 into a brief summary of the regression models, showing the overall fit and significant predictors. To maintain full scientific transparency, the detailed output for all five regression models has been transferred to the Supplementary Materials (Table S7). The accompanying text has been rewritten to provide a more focused and thematic summary of the key findings, including an interpretation of the relative strength of the predictors.

Reviewer 3 Report

Comments and Suggestions for Authors

The paper by Ispas GA describes how parental weight stigma, internalized bias, and anti-fat attitudes relate to different forms of parent-child communication about weight and health. It has been conducted in Romania, a context that is often underrepresented in obesity and stigma research. The reading of the paper suggests some comments:

1) The structure of the Introduction needs improvement. It currently opens by stating the study’s aims and methods (lines 46-50) before presenting the theoretical background and rationale. These aims are then repeated again at the end (lines 122–138), which creates confusion and redundancy. Please reorganize the Introduction to follow a logical and standard flow: General background, review of the literature with identified knowledge gaps, justification for studying this topic in Romania, clear statement of aims and hypotheses (only once, at the end of the Introduction).

2) The literature review on parent-child communication (lines 74-95) is longer than necessary and contains overlapping points (lines 101-110 contain repetitions). This section could be simplified to focus more clearly on what is not yet known.

3) Please, insert a flow chart related to the included population.

4) Please explain more clearly how participants were recruited. Was convenience sampling used? Were invitations distributed through schools or social media? This is essential for understanding sample representativeness.

5) No information is given about whether a power analysis was conducted to justify the sample size. Please consider including this.

6) The sample is heavily unbalanced, with 91.8% female participants and 78.5% holding tertiary education. These details should be reported more clearly and discussed later in the limitations. Furthermore, this unbalance could have influenced the results.

7) Anthropometric measures (weight and height) for both parents and children were self-reported. While this is acknowledged briefly, a more detailed discussion of the potential for bias is needed.

8) The focus on weight-related conversations dominates the results and discussion, while findings related to health-focused communication are covered less thoroughly. Please consider balancing these sections.

9) Several statistically significant findings show very small effect sizes, for example in models predicting comments about others' weight or diet. These should be interpreted with more caution. Maybe could be useful emphasize the practical relevance of results, not just statistical significance in the discussion.

10) Post hoc analyses: The Games-Howell test is used appropriately, but no explanation is provided for readers unfamiliar with it. A short clarification when it first appears would help improve accessibility.

11) Discussion. Although Section 4.4 is titled “Limitations,” it is too brief and misses several key points. Please expand this section to include: Self-reported height and weight data (risk of misreporting), Potential for social desirability bias, especially among highly educated parents, A non-representative sample in terms of gender, education, and geography, The use of only parent-reported outcomes, without validation from the child’s perspective or observed behavior,

12) The conclusions are sometimes too strong or prescriptive for a cross-sectional, observational study (e.g., “Interventions should…”). Please rephrase using more cautious or tentative language (e.g., “might benefit from,” “could consider…”).

13) While implications are discussed, they are not clearly linked to real-world applications such as schools or healthcare settings. Consider including examples and suggestions for future research, such as longitudinal studies, child-reported outcomes, or intervention trials.

14) Please, delete from discussion the repetition of numbers of results. Maintain only the concepts.

Author Response

Comment 1: "The structure of the Introduction needs improvement. It currently opens by stating the study’s aims and methods (lines 46-50) before presenting the theoretical background and rationale. These aims are then repeated again at the end (lines 122–138), which creates confusion and redundancy. Please reorganize the Introduction to follow a logical and standard flow: General background, review of the literature with identified knowledge gaps, justification for studying this topic in Romania, clear statement of aims and hypotheses (only once, at the end of the Introduction)."

Response 1: Thank you for this excellent and constructive suggestion. We have completely restructured the Introduction to follow the logical and standard scientific flow you recommended. The section now begins with the general background and prevalence data, transitions to a review of the relevant literature, highlights the specific knowledge gaps, and provides a clear justification for the study within the Romanian context. As requested, the detailed study objectives are now presented only once, in the final paragraph, to eliminate redundancy and clearly state the study's aims.

Comment 2: "The literature review on parent-child communication (lines 74-95) is longer than necessary and contains overlapping points (lines 101-110 contain repetitions). This section could be simplified to focus more clearly on what is not yet known."

Response 2: Thank you for pointing out the need for greater conciseness in our literature review. We have revised and condensed the paragraphs discussing parent-child communication, removing repetitive statements and streamlining the text. The updated section now more efficiently contrasts weight-focused and health-focused communication styles and their associated outcomes, leading more directly into the identified gaps in the research. We believe this revision makes the rationale for our study clearer and more impactful.

Comment 3: "Please, insert a flow chart related to the included population"

Response 3: We have created a participant flow diagram as requested, which will be included as Figure 1 (Line 317). The text has been updated to reference this figure, which visually details the progression from the initial number of survey respondents to the final analytic sample, including the number of participants excluded at each stage and the reasons for exclusion.

Comment 4: "Please explain more clearly how participants were recruited. Was convenience sampling used? Were invitations distributed through schools or social media? This is essential for understanding sample representativeness."

Response 4: Thank you for the request for clarification. We have revised the text to be more explicit about our recruitment strategy. The manuscript now states that a non-probability convenience sampling method was used, combining online advertisements and community outreach through partner schools to recruit a diverse sample of parents from the specified region. 

Comment 5: "No information is given about whether a power analysis was conducted to justify the sample size. Please consider including this."

Response 5: Thank you for raising this critical point. We have clarified in the manuscript that an a priori sample size calculation was performed before recruitment to ensure the study would be adequately powered for our objectives. Based on the population of school-aged children in Cluj County (176,858), it was determined that a minimum sample of 384 participants was required for a 95% confidence level and a 5% margin of error. Our final analytic sample of N=414 successfully meets this requirement, and the text has been updated to reflect this planning.

Comment 6: "The sample is heavily unbalanced, with 91.8% female participants and 78.5% holding tertiary education. These details should be reported more clearly and discussed later in the limitations. Furthermore, this unbalance could have influenced the results."

Response 6: We thank the reviewer for pointing out the need to articulate the sample's demographic characteristics and their implications more clearly. We have revised the limitations section to explicitly state the high percentages of female and highly educated participants. Furthermore, we now acknowledge directly that this significant demographic imbalance limits the generalizability of our findings and could have influenced the observed results.

Comment 7: "Anthropometric measures (weight and height) for both parents and children were self-reported. While this is acknowledged briefly, a more detailed discussion of the potential for bias is needed."

Response 7: We thank the reviewer for this feedback. We acknowledge that our initial discussion of this limitation was too brief. We have now expanded the paragraph addressing self-reported anthropometric data to provide a more detailed account of the potential for systematic reporting bias (e.g., underestimation of weight) and to elaborate on how such measurement error could have influenced the observed associations.

Comment 8: "The focus on weight-related conversations dominates the results and discussion, while findings related to health-focused communication are covered less thoroughly. Please consider balancing these sections."

Response 8: This is an excellent point, and we appreciate the opportunity to provide a more balanced interpretation. We have added a new paragraph to the Discussion dedicated to the findings on general health-focused communication. This new text contrasts the predictors of general health talk with those of weight-specific talk, highlighting that the former was associated with sociodemographic factors like education rather than the psychosocial factors driving weight talk.

Comment 9: "Several statistically significant findings show very small effect sizes, for example in models predicting comments about others' weight or diet. These should be interpreted with more caution. Maybe could be useful emphasize the practical relevance of results, not just statistical significance in the discussion."

Response 9: We thank the reviewer for this important critique regarding the interpretation of our findings. We agree that this is a critical point to address. We have added a new paragraph to the limitations section that explicitly cautions against overinterpreting statistical significance in the context of small effect sizes and emphasizes the need to consider the practical relevance of these findings. 

Comment 10: "Post hoc analyses: The Games-Howell test is used appropriately, but no explanation is provided for readers unfamiliar with it. A short clarification when it first appears would help improve accessibility."

Response 10: Thank you for this excellent suggestion to improve the manuscript's accessibility. We have now added a brief explanatory phrase where the Games-Howell test is first mentioned, clarifying that it is a post-hoc test appropriate for comparing groups when the assumption of equal variances is not met.

Comment 11: "Discussion. Although Section 4.4 is titled “Limitations,” it is too brief and misses several key points. Please expand this section to include: Self-reported height and weight data (risk of misreporting), Potential for social desirability bias, especially among highly educated parents, A non-representative sample in terms of gender, education, and geography, The use of only parent-reported outcomes, without validation from the child’s perspective or observed behavior."

Response 11: We thank the reviewer for these specific and constructive recommendations. We have thoroughly revised and expanded the limitations section to incorporate each of these key points. The revised section now includes a more detailed discussion of the potential for misreporting with self-reported anthropometric data; an explicit acknowledgment of social desirability bias, particularly relevant to our highly educated sample; a more detailed description of the sample's lack of representativeness in terms of gender, education, and geography; and a new point on the limitation of using only parent-reported outcomes without validation from the child or through objective observation.

Comment 12: "The conclusions are sometimes too strong or prescriptive for a cross-sectional, observational study (e.g., “Interventions should…”). Please rephrase using more cautious or tentative language (e.g., “might benefit from,” “could consider…”).

Response 12: We thank the reviewer for this important reminder regarding the interpretation of our findings. We agree that the language in the conclusions may have sounded too prescriptive for a cross-sectional study. We have revised this section to use more cautious and tentative phrasing (e.g., "suggest that... could be," "might help foster," "may represent a valuable opportunity") to ensure our conclusions accurately reflect the correlational nature of the data."

Comment 13: "While implications are discussed, they are not clearly linked to real-world applications such as schools or healthcare settings. Consider including examples and suggestions for future research, such as longitudinal studies, child-reported outcomes, or intervention trials."

Response 13: We thank the reviewer for these excellent and specific suggestions to strengthen the manuscript. We have revised the "Implications for Practice" section to include more concrete examples of real-world applications, expanding beyond healthcare settings to include the role of schools and public health initiatives. In the "Future Research" section, we have reinforced the call for longitudinal studies and the inclusion of child-reported outcomes, and we have added an explicit recommendation for the design and implementation of intervention trials to test the effectiveness of programs targeting parental bias and communication.

Comment 14: "Please, delete from discussion the repetition of numbers of results. Maintain only the concepts."

Response 14: We appreciate this valuable suggestion to improve the narrative flow of the Discussion. We have revised the entire section to remove specific statistical values (such as correlation coefficients, beta values, and confidence intervals). The text now focuses on presenting the conceptual meaning and implications of the findings, using descriptive language (e.g., “a strong association,” “a significant predictor”) to convey the results’ magnitude and importance.